# The Effect and Regulatory Mechanism of High Mobility Group Box-1 Protein on Immune Cells in Inflammatory Diseases

**DOI:** 10.3390/cells10051044

**Published:** 2021-04-28

**Authors:** Yun Ge, Man Huang, Yong-ming Yao

**Affiliations:** 1Department of General Intensive Care Unit, Second Affiliated Hospital of Zhejiang University School of Medicine, Hangzhou 310052, China; 20130505@zju.edu.cn; 2Translational Medicine Research Center, Medical Innovation Research Division and Fourth Medical Center of the Chinese PLA General Hospital, Beijing 100048, China

**Keywords:** high mobility group box-1 protein, immune cells, inflammatory disorders, damage-associated molecular pattern

## Abstract

High mobility group box-1 protein (HMGB1), a member of the high mobility group protein superfamily, is an abundant and ubiquitously expressed nuclear protein. Intracellular HMGB1 is released by immune and necrotic cells and secreted HMGB1 activates a range of immune cells, contributing to the excessive release of inflammatory cytokines and promoting processes such as cell migration and adhesion. Moreover, HMGB1 is a typical damage-associated molecular pattern molecule that participates in various inflammatory and immune responses. In these ways, it plays a critical role in the pathophysiology of inflammatory diseases. Herein, we review the effects of HMGB1 on various immune cell types and describe the molecular mechanisms by which it contributes to the development of inflammatory disorders. Finally, we address the therapeutic potential of targeting HMGB1.

## 1. Introduction

Living organisms have evolved alarm systems that can sense cellular stress and damage, facilitating the discharge of several critical endogenous factors such as damage-associated molecular pattern molecules (DAMPs) and alarmins [1,2,3,4]. These molecules maintain homeostasis by acting intracellularly to elicit immune and inflammatory responses [5,6,7]. However, they also contribute to tissue damage and organ dysfunction and are therefore associated with the pathogenesis of various inflammatory disorders [8]. Targeting these molecules may constitute a promising therapeutic approach against these diseases [9,10,11,12].

High mobility group box-1 protein (HMGB1), a prototypical alarmin and DAMP molecule, was first described in 1973 [13,14]. The protein was isolated from calf thymus chromatin and was classified as belonging to a “high-mobility group” due to its rapid migration in polyacrylamide gel electrophoresis [14]. In the 1990s, the DNA-binding region of the HMGB1 protein, called the HMG box, was identified. Nuclear HMGB1 was found to bind DNA and to serve as a chromatin-associated factor that maintains the stability and structure of chromosomes [15,16,17,18].

Almost a decade later, the receptor for advanced glycation end products (RAGE) was found to act as a special receptor for HMGB1, helping to explain the functions of extracellular HMGB1 [19,20,21,22]. HMGB1 also binds Toll-like receptor (TLR) 4 to elicit inflammatory responses [23,24] and it interacts with a variety of extracellular partners that undergo endocytosis and enter the endolysosomal system [25,26,27].

Itis evident that HMGB1 plays a cardinal role in regulating inflammatory responses and in cell fate decisions [28,29,30]. Importantly, HMGB1 has been implicated in the pathogenesis of numerous inflammatory disorders including sepsis, lung conditions, autoimmune diseases, acute liver injury, cardiac injury, encephalopathy and other inflammation-driven conditions [8,31,32,33,34,35].

This review provides an update in the HMGB1 field, with focus on the pathologic role of HMGB1 in inflammatory and immune conditions. It also highlights the evidence supporting the potential of HMGB1 as a therapeutic target.

## 2. HMGB1 Characteristics

### 2.1. The HMGB1 Protein

The *HMGB1* gene, which was sequenced in 1996, is located on chromosome 13q12.3. It encodes a nuclear protein of 215 amino acids with a molecular weight of 25 kDa [36,37,38]. The HMGB1 transcriptome comprises three isoforms: sulfonyl HMGB1, disulfide HMGB1 and fully reduced HMGB1 [39]. Disulfide HMGB1 is the only isoform with proinflammatory cytokine-like properties. HMGB1 shows high sequence homology across species, including fish, plants, bacteria, *Chironomidae*, *Drosophila* and yeast [40,41]. Rodent and human HMGB1 share 99% homology, while the rat and mouse homologues have identical amino acid sequences [40]. The DNA-binding domain of HMGB1 comprises an A-box domain (residues 9–79) and B-box domain (residues 95–163), separated by a short linker peptide (Figure 1). The C-terminal domain, which includes a highly acidic tail, is required to recruit p53 to bind its target DNA and thereby modulate cell cycle and death pathways [41].

### 2.2. Regulation of HMGB1 and Its Release Into the Extracellular Space

HMGB1 is usually located in the nucleus due to its two unique nuclear localization signals (NLS), NLS1 (residues 28–44) and NLS2 (residues 179–185) [42,43]. Deacetylation and acetylation of NLS1 or NLS2 lead to rapid shuttling from the nucleus to the cytoplasm. Phosphorylation of serine residues within NLS1 and NLS2 may also stimulate nucleocytoplasmic diffusion of HMGB1 [44,45]. Following phosphorylation of HMGB1, it binds to karyopherin-α1 and thereby remains sequestered in the cytoplasm. Protein kinase C (PKC) regulates the phosphorylation of HMGB1: levels of cytoplasmic HMGB1 fall when PKC is inhibited and rise when PKC activity increases [46,47].

Methylation of HMGB1 dampens the strength of HMGB1-DNA binding, leading to its translocation out of the nucleus. This explains the observed release of HMGB1 from neutrophils under chronic inflammation [48,49,50]. Cytoplasmic translocation of HMGB1 can also be induced by cellular stress that triggers posttranslational modifications, i.e., oxidation, methylation, phosphorylation, or hyperacetylation [44,45,46,47]. These modifications accelerate secretion of HMGB1 from the cell into the extracellular environment through secretory lysosomes. Hyperacetylation of HMGB1 has been observed in cells and animals subjected to oxidative stress [46]. On exposure to such stress, HMGB1 forms a complex that includes a nuclear export factor and chromosome region maintenance-1 (CRM-1) protein, resulting in the shuttling of HMGB1. Three cysteine residues in HMGB1 (Cys106, Cys45 and Cys23 in the human protein) regulate its nucleocytoplasmic translocation under oxidative stress [51,52,53,54]. Exposing macrophages to lipopolysaccharide (LPS) leads to intracellular production of hydrogen peroxide, which in turn leads to the formation of an intramolecular disulfide between Cys45 and Cys23 [55]. This results in transfer of HMGB1 from the nucleus to the cytoplasm and subsequent secretion from macrophages.

In addition to the ability of stress and posttranslational modifications to promote HMGB1 secretion, the protein can also passively diffuse out of leaking necrotic cells to induce inflammation. In fact, HMGB1 released from necrotic cells can promote inflammation more strongly and persistently than HMGB1 released from apoptotic cells [56,57,58].

## 3. HMGB1 Receptor Networks

At least 14 receptors have been identified to interact with extracellular HMGB1 (Figure 2), with RAGE and TLR4 perhaps the most extensively explored [59,60]. Whether some of the other apparent HMGB1 receptors are specific for HMGB1 is unclear, because HMGB1 can be modified post-translationally and it can interact with numerous immune-related mediators, including interleukin (IL)-1α, IL-1β, LPS, nucleosomes, histones, RNA, DNA and SDF-1 [61,62,63,64]. This molecular cooperation is essential for HMGB1-induced inflammatory responses and the cooperative mechanisms are discussed later in the review.

RAGE was first described in 1995 as the amphoterin-binding receptor [20]. A member of the immunoglobulin superfamily, RAGE was originally suggested to play a role in neurite outgrowth. The protein contains a 43-amino acid cytoplasmic tail, a short transmembrane domain and an extracellular region [61,62]. Its extracellular region is required for ligand binding, while the cytoplasmic tail is responsible for signal transduction. RAGE binds various molecules, such as HMGB1, RNA, DNA, amyloid-β peptide and S100 family members [26,63]. The RAGE binding domain in HMGB1 is located at residues 150-183 [61] and binding of RAGE to HMGB1 in macrophages triggers HMGB1 endocytosis and cell pyroptosis, leading to the production of proinflammatory cytokines [6]. This work may explain the synergy observed when the immune system detects proinflammatory molecules complexed with HMGB1.

Both TLR4 and RAGE are required for HMGB1 to trigger the release of mediators from macrophages. In macrophages lacking RAGE, their activation via TLR4 leads to much lower cytokine secretion [64]. HMGB1 can also prime pyroptosis to promote cytokine synthesis and endosome-mediated secretion [6,65].

A second potential binding site for RAGE was identified in HMGB1, at residues 23–50 in the A-box [66]. While this remains to be confirmed, the presence of multiple RAGE binding sites may help explain how RAGE binding can give rise to the diverse functions of HMGB1 in synergy with its partner mediators. Accumulating evidence suggests that HMGB1-RAGE binding is associated with a diverse array of inflammatory disorders, including sepsis, atherosclerosis, rheumatoid arthritis, neurological diseases and diabetic nephropathy [8,67].

Toll-like receptors (TLRs) can recognize various danger signals, such as DAMPs and pathogen-associated molecular patterns (PAMPs) to trigger immune responses against pathogens [68,69]. Binding of HMGB1 to TLR2, TLR4 and TLR9 initiates nuclear factor-κB (NF-κB) signaling as well as the release of chemokines and proinflammatory cytokines. The interaction between HMGB1 and TLR2 can activate natural killer cells and stem cells via induction of Smads, STATs and NF-κB signaling [70,71]. The interaction of HMGB1 with TLR4 and its coreceptor MD-2 is essential for mediator release from macrophages. MD-2 interacts with disulfide HMGB1. In macrophages, knockdown of MD-2 reduces NF-κB translocation and production of tumor necrosis factor (TNF) in response to HMGB1 [72]. HMGB1-TLR4 signaling can also stimulate anticancer immunity by counteracting cell adhesion, angiogenesis and migration [73].

In addition to RAGE and TLRs, TIM-3 can bind with HMGB1. This interaction inhibits the recruitment of nucleic acids into dendritic cell endosomes [59,60]. Thus, blocking TIM-3 can potentiate chemotherapy and DNA vaccination by improving the immunogenicity of nucleic acids released from dying tumor cells [74]. The interaction of HMGB1 with N-methyl-D-aspartate receptor (NMDAR) can activate NMDAR-induced cell responses under cellular stress [75]. Binding of HMGB1 to CD24/Siglec-10 receptor suppresses HMGB1-mediated NF-κB signaling and following release of pro-inflammatory molecules [76]. Interestingly, the recently identified triggering receptor expressed on myeloid cell-1 engages in HMGB1-dependent NF-κB activation, resulting in the release of pro-inflammatory mediators during sepsis progression [77,78].

## 4. Effects of HMGB1 on Immune Cell Types and the Regulatory Mechanisms Involved

HMGB1 is produced by various immune cell types, especially neutrophils and monocytes, after stimulation with IL-1, TNF-α, DNA, CpG and LPS [43,48,49,50,58]. HMGB1 acts as an inflammatory molecule: it accelerates the release of proinflammatory cytokines to elicit a range of immunological processes [28,79,80,81,82,83] (Figure 3). At the same time, HMGB1 can induce systemic inflammatory responses such as fever, arthritis, acute lung injury, anorexia and weight loss [8].

Binding of extracellular HMGB1 to TLRs and RAGE on the surface of immune cells initiates downstream signaling cascades involving extracellular signal-regulated kinase 1/2 (ERK1/2), p38 mitogen-activated protein kinase (MAPK), phosphoinositide-3-kinase (PI3K)/RAC-α serine/threonine-protein kinase (Akt) and NF-κB (p56) [24,65,81,84,85,86,87]. These pathways activate immune cells and promote the release of inflammatory mediators that induce cell migration, adhesion, proliferation and angiogenesis [6,24,54,74,88].

### 4.1. HMGB1 and Neutrophils

Neutrophils are the first responders of innate immunity; they produce reactive oxygen species (ROS), cytokines, antibacterial peptides and other inflammatory molecules. In response to infection, neutrophils generated by innate immune cells are recruited to the site of infection and engage pathogens [89]. Once within the tissues, neutrophils phagocytose pathogens and they produce a series of proteins that possess antimicrobial properties and can remodel tissue. However, excessive neutrophil activation leads to degranulation and release of ROS, resulting in host tissue damage [89,90].

The interaction between HMGB1 and RAGE stimulates the recruitment and infiltration of neutrophils into necrotic sites, which can exacerbate hepatic injury [32,91,92]. Similarly, HMGB1 can injure glomerular endothelial cells by amplifying neutrophil activation in patients with vasculitis involving the anti-neutrophil cytoplasmic antibody [93]. HMGB1/TLR4 signaling promotes neutrophil migration and contributes to paraquat-induced acute lung injury [94]. Therefore, HMGB1 acts as a key amplifier of neutrophil-driven tissue necrosis and organ injury, making it a potential therapeutic target.

The recent discovery of neutrophil extracellular traps (NETs) has expanded our view of how neutrophils function [95]. NETs are secreted from neutrophils and aid in pathogen clearance at the site of inflammation [96,97]. However, extensive formation of NETs exacerbates damaging inflammatory responses and tissue injury. Accumulating evidence indicates that HMGB1 can be released from NETs and the interaction of HMGB1 with TLR4 can accelerate NET formation [98], which may be further potentiated in the presence of IgG against anti-neutrophil cytoplasmic antibody [93,99]. HMGB1 appears to act via TLR2, TLR4, RAGE and NADPH oxidase to accelerate NET formation [24,59,94]. In addition, NET-derived HMGB1 can act via RAGE-dynamin cascades to induce macrophage pyroptosis, which aggravates inflammation [100].

Macrophages normally phagocytose apoptotic neutrophils and NETs in a process termed NETosis and impairment in this NET clearance may lead to persistent inflammatory responses and subsequent organ injury [24,101]. NETosis is markedly decreased in patients with acute respiratory distress syndrome [101] and impaired NETosis has been associated with worse liver damage following sepsis and neuronal impairment in the ischemic brain [102]. HMGB1 strongly promotes neutrophil activation and NET release, contributing to inflammatory disorders. Inhibiting HMGB1 or NET formation may mitigate pulmonary inflammation in acute respiratory distress syndrome [101] and help treat inflammatory diseases such as diabetic wounds and COVID-19 [103,104].

### 4.2. HMGB1 and Macrophages

Monocytes are not very abundant in the peripheral circulation. They are produced in the bone marrow and released into the peripheral circulation, where they circulate for only about a day before settling permanently within tissues [105]. Once settled, the cells are called tissue macrophages. Macrophages are widely distributed in lymphoid and non-lymphoid tissues and, because of their prodigious phagocytic ability, they are critical for presenting antigens from particulate immunogens such as bacteria [105,106].

On exposure to LPS, macrophages produced hydrogen peroxide that promoted the formation of intramolecular disulfide between C45 and C23 [107]. This oxidative response initiated HMGB1 release from macrophages [77]. Extracellular HMGB1 is recognized as a danger signal that provokes proinflammatory responses and impairs efferocytosis and phagocytosis of macrophages [101,108,109]. Administering an HMGB1 antagonist or anti-HMGB1 antibody to cultured macrophages suppresses internalization of HMGB1 and HMGB1-LPS complexes, blocking macrophage activation and thereby inhibiting the inflammatory response [110,111,112,113,114,115].

Macrophages are implicated in initiating, maintaining and resolving inflammatory processes in host defense and during sepsis [100,116]. C1q is a regulator of inflammation that interacts with DAMPs and maintains monocytes in a quiescent state in order to promote anti-inflammatory macrophages [117]. HMGB1 together with C1q stimulate an anti-inflammatory macrophage response, but suppress macrophage plasticity. The HMGB1-C1q complexes modulate macrophage activities by switching between specialized pro-resolving molecules such as biosynthetic enzymes and leukotriene [118]. Circulating HMGB1 acts on macrophages as an inflammatory cytokine in later stages of sepsis. HMGB1 acts via Janus kinase (JAK) and activator of transcription (STAT) pathways to trigger production of a broad range of chemokines and inflammatory molecules in macrophages [119]. The HMGB1-RAGE interaction mediates intracellular cascades involving NF-κB and other products of activated macrophages [106].

HMGB1 has been implicated in various forms of programmed cell death. For example, HMGB1 induces apoptosis of mouse macrophages in a dose- and time-dependent manner [120,121]. In sepsis, widespread macrophage apoptosis causes immune dysfunction and even immune paralysis. HMGB1 stimulation of macrophages strongly activates caspase-3 [120,121,122], which initiates cell toxicity and programmed cell death. HMGB1 also initiates an endoplasmic reticulum stress response to induce the formation of macrophage-derived foam cells as well as their apoptosis [123]. HMGB1 is also involved in another type of programmed cell death called pyroptosis, characterized by plasma rupture, DNA fragmentation and production of proinflammatory mediators. Following endocytosis, NET-derived HMGB1 may trigger intra-macrophage-induced pyroptosis, which in turn may exacerbate inflammation [65,100,124,125,126]. Reciprocally, alveolar macrophage pyroptosis induced by NOD-like receptor family pyrin domain-containing protein 3 promotes HMGB1 production in acute lung injury [65]. Poly (ADP-ribosylated) HMGB1 prevents macrophages from phagocytosing apoptotic cells, preventing the normal resolution of inflammation [92,101,108] and, in fact, this effect of HMGB1 is much weaker when the HMGB1 is unmodified, suggesting a therapeutic strategy to diminish inflammation.

### 4.3. HMGB1 and Dendritic Cells

Dendritic cells (DCs) serve as professional antigen-presenting cells that capture antigen in one location, then migrate to lymph nodes, where they present it to native T cells, initiating the adaptive immune response [127]. In addition to presenting foreign antigens, DCs stimulate the proliferation of T lymphocytes and regulatory T cells, which depends on the ratio of mature to immature DCs [128]. Normally, the immature DCs express low amounts of MHCII, CD86, CD80, CD11c, but high amounts of CD45RB. In a mouse model of severe trauma, administration of CD11c^low^CD45RN^high^ DCs prevented acute inflammatory responses by inhibiting the formation of pro-inflammatory mediators [129].

HMGB1 finely tunes the maturation, differentiation and immune functions of DCs, so it influences the shift of helper T (Th) 1 cells and Th17 polarization necessary for T cell-mediated immunity [127,128,129,130]. HMGB1 is critical for CXCL12 activity, which attracts myeloid-derived cells, thereby promoting recruitment and motility of leukocytes [119]. The Hp91 sequence within the B-Box domain of HMGB1 is required for DC activation; via this domain, HMGB1 enters DCs in a clathrin- and dynamin-dependent manner [131]. Hp91-mediated DC activation is dependent on TLR4, MyD88 and IFNαβR and it is mediated by NF-κB and p38 MAPK cascades [131,132].

Several studies suggest that HMGB1 acts as a diphasic immune regulator of DC function. Our own studies revealed that HMGB1 can stimulate DC apoptosis in a time- and dose-dependent manner [133,134]. Conversely, blockade of HMGB1 in burn tissue was found to increase the expression of DC costimulatory molecules including MHC, CD86 and CD80 as well as IL-2 [97]. The abundant production of HMGB1 may induce the maturation of DCs and Th2 polarization, which would suppress T cell-mediated immunity. For example, we found that HMGB1 can facilitate the differentiation of DC cells into the CD11c^low^CD45RN^high^ subtype, leading to a reduction in T cell-activated immune responses [133,134].

In DCs, HMGB1 upregulates PI3K, Akt and mTOR and phosphorylated proteins [135]. The HMGB1/PI3K/Akt/mTOR axis promotes adhesion, maturation, chemotactic and antigen-presenting ability of lung DCs, implicating it in lung inflammation [135]. Similarly, HMGB1 promotes IL-9 release to activate group 2 innate lymphoid cells and DCs, aggravating asthma [128]. In asthma, HMGB1 also contributes to hyper-responsiveness and airway inflammation via the pathway involving adenosine triphosphate (ATP)/purinergic receptor P2X ligand-gated ion channel 7 [135].

HMGB1-dependent DCs activation induces Th17-type responses and this activation can be blocked using a soluble form of RAGE (sRAGE) [136], which blocks interaction between HMGB1 and endogenous RAGE. DCs pretreated with recombinant HMGB1 and sRAGE suppressed Th17-dependent cytokine release, eliminating neutrophil airway inflammation [136,137]. Together, these studies suggest that neutralizing HMGB1 may mitigate inflammatory responses.

HMGB1 upregulates CD86 and CD80 levels on mucosal DCs, whereas the HMGB1 inhibitor glycyrrhizin inhibits induction of DCs and activation of CD8+ cytotoxic lymphocytes that specifically recognize intestinal ovalbumin [131,138]. 

The pivotal role of HMGB1 in DC activities may make it a target against human immunodeficiency virus (HIV) infection. HIV can reside within DCs and thereby spread and evade host immunity [139,140]. Neutralization of HMGB1 might render HIV-infected DCs vulnerable to natural killer cells.

### 4.4. HMGB1 and T Lymphocytes

T lymphocytes are the principal players in cell-mediated adaptive immunity. T lymphocytes are divided into two major cell types: Th cells and T cytotoxic cells [141]. Th1 cells regulate the immune response to intracellular pathogens and Th2 cells modulate the response to many extracellular pathogens. The cell activity is remarkably dampened that drives the response in favor of Th2 profile. A Th2-dominant immune response or excessive apoptosis of T lymphocytes may render the host more vulnerable to infection [141].

HMGB1 potently modulates T cell immunological reactions, but the underlying molecular mechanisms remain unknown. We observed that serum HMGB1 was markedly increased in rats following thermal injury and it was able to activate T lymphocytes at relatively low concentrations [142]. HMGB1 at low concentrations enhanced the proportions of Th17 and CD4+ T cells and it increased the CD4+/CD8+ ratio, which might aggravate inflammatory damage. HMGB1 at high concentrations, conversely, suppressed T lymphocyte activity, highlighting the dual effects of HMGB1 on T lymphocytes. Inhibition of HMGB1 restored normal levels of T cell proliferation and IL-2 and it shifted to a Th1 profile.

IL-2 potently stimulates T lymphocytes and regulates the balance between Th1 and Th2 cells. HMGB1 exerts its immunosuppressive effect on T lymphocytes via p53, p38 MAPK and ERK 1/2 [143,144,145,146,147,148,149]. p38 MAPK and ERK1/2 may act on the transcription factor NF-AT in HMGB1-induced immunosuppression. NF-AT can also interact with the IL-2 promoter and modulate the transcription of the IL-2 gene [142]. Thus, HMGB1 may operate as an activator of NF-AT expression and IL-2 secretion. At the same time, excessive release of HMGB1 can reduce IL-2 levels, suggestive of HMGB1′s immunosuppressive properties [142,143].

The proinflammatory B-box of HMGB1 is required for soluble CD52 to be able to suppress T cells. CD52 is a glycophosphatidylinositol (GPI)-anchored glycoprotein expressed by T cells, B cells, DCs, natural killer cells, macrophages and eosinophils [143,144,150]. 

HMGB1 can directly evoke the differentiation of Th17 and Th2 cells, through which the protein promotes mucus production and airway inflammation in asthmatic mice [148]. After injury, release of HMGB1 immunosuppresses T lymphocytes [151,152] and we found that stimulation with HMGB1 can induce late apoptosis and necrosis as well as mitochondrial apoptosis in T cells [151]. We also found that antibody neutralization of HMGB1 promoted the T cell-dependent immune response in septic rats, thereby ameliorating multiple organ damage [153,154]. Similarly, our studies showed that mitofusin-2 can reverse HMGB1-induced immunosuppression of T cells [151,152]. 

### 4.5. HMGB1 and Regulatory T Cells

Regulatory T cells are a subset of CD4^+^ T cells that dampen the proliferation and activity of antigen-presenting cells, T cells, natural killer cells and B cells [155,156]. These cells attenuate the immune response by constitutively expressing intracellular cytotoxic T lymphocyte-associated antigen-4 (CTLA-4), glucocorticoid-induced TNF receptor (GITR) and forkhead/winged helix transcription factor p3 (Foxp3), as well as by producing the immunosuppressive cytokines IL-10 and transforming growth factor (TGF)-β. Therefore, regulatory T cells regulate both humoral and cell-mediated immunity [155,156].

In regulatory T cells from mouse spleen, we noted that HMGB1 markedly down-regulated expressions of Foxp3 and CTLA-4, which was accompanied by the production of IL-10 [157,158]. Another study showed that HMGB1 suppressed regulatory T cells via a RAGE-driven pathway and limited the activity of CD4+CD25-CD127+ conventional T cells. HMGB1 can induce migration and survival of regulatory T cells [156]. In rat model of burn injury, anti-RAGE antibody led to a profound decrease in the immune functions of regulatory T cells, while restoring effector T cell activity [157]. Therefore, following burn injury, HMGB1 binding to RAGE may suppress regulatory T cell activity, thereby activating effector T cells.

TLRs are sensors that recognize invading pathogens and mediate signaling cascades that trigger and regulate immune responses. The binding of TLR4 with HMGB1 regulates the function of regulatory T cells. We found that stimulation of regulatory T cells with HMGB1 led to a massive elevation of cytoplasmic TLR4 but a steep decline in membrane TLR4 [159]. These changes were reversed by an anti-TLR4 neutralizing antibody. The interaction between HMGB1 and TLR4 may also suppress regulatory T cells via NF-κB signaling [159].

Regulatory T cells are critical for resolving lung damage. Inhibiting HMGB1 dampens pro-inflammatory cytokine production, increases TGF-β release and attenuates lung injury [160]. An HMGB1/PTEN/β-catenin cascade modulates the development and involvement of regulatory T cells in sepsis-associated lung injury. Elevated HMGB1 may drive a Th17 response and suppress regulatory T cells during pulmonary inflammation [160,161,162].

## 5. Role of HMGB1 in Various Inflammatory Disorders

The activities of HMGB1 depend greatly on its cellular localization. Nuclear HMGB1 acts as a pro-inflammatory molecule in sterile tissue injury or during infection, which in turn affects immune responses. Extracellular HMGB1 engages in various activities affecting inflammation, oxidation, migration, invasion, proliferation, differentiation and tissue regeneration. For example, HMGB1 induces cultured human monocytes to produce a plethora of proinflammatory mediators such as macrophage inflammatory protein (MIP)-1, IL-1, IL-6, IL-8 and TNF [29,31]. In vivo, HMGB1 can initiate systemic inflammation that can be associated with fever, epithelial barrier dysfunction, endothelial cell activation, acute lung injury, anemia, cognitive dysfunction, arthritis, anorexia and even death [77,78,163]. Thus, many studies suggest that HMGB1 possesses fulminant inflammatory properties and contributes to numerous inflammatory disorders (Figure 4, Table 1).

### 5.1. Sepsis

HMGB1 appears much later than other proinflammatory cytokines after onset of sepsis: LPS stimulation of mouse macrophages upregulated HMGB1 within 8 h, followed by a massive increase at 16–32 h [77,78].

HMGB1 can inspect and carry immunogenic nucleic acids, upregulating IL-6 and type 1 interferons in immune cells [164]. Binding of extracellular HMGB1 to viral nucleic acids leads to their internalization via dynamin-dependent endocytosis, which leads in turn to cytokine and interferon responses [164,165]. Silencing of HMGB1 in cells stimulated with viral nucleic acids dampened the immune response substantially [164,165]. These studies indicate that HMGB1 serves as a viral sentinel in a nucleic acid-dependent manner.

The role of HMGB1 in sepsis has been extensively explored in rodent models with cecal ligation and puncture (CLP). The fact that HMGB1 acts late in sepsis opens a therapeutic window for medical treatment [166,167,168,169,170,171]. Silencing HMGB1 in a mouse model of sepsis mitigated the cytokine storm of DCs and macrophages, as well as reduced lymphocyte apoptosis and mortality [120]. Similarly, targeting HMGB1 with a specific antibody promoted neutrophil activity, attenuated post-sepsis immunosuppression and conferred resistance to secondary bacterial infection [172]. Targeting HMGB1 has shown promise in preclinical studies, but it has not yet been tested in clinical trials.

### 5.2. Autoimmune Diseases

Autoimmune diseases are attributed to dysregulation of both immune and inflammatory responses. Recent findings suggest that HMGB1 is critical for the development of autoimmune pathologies [173].

Rheumatoid arthritis (RA) involves destructive synovitis at the cartilage-bone interface, along with synovial tissue hypoxia [173,174]. HMGB1 is upregulated in damaged pannus tissue from RA patients [175,176] and in a mouse model of RA [173]. In addition, administration of HMGB1 to mice leads to destructive arthritis [177,178,179]. HMGB1 in patients with RA synovitis may be produced by vascular endothelial cells, fibroblasts and activated synovial macrophages and the form of HMGB1 in the synovial fluid is hyperacetylated [174]. Moreover, extracellular HMGB1 released from dying hypoxic cells can trigger the secretion of proinflammatory cytokines such as IL-1 and TNF [171]. Blockade of HMGB1 in animal models of RA ameliorates bone and cartilage lesions [114,115,180,181,182,183,184].

Systemic lupus erythematosus (SLE) is a systemic inflammatory and autoimmune disorder characterized by the deposition of immune complexes in multiple organs [185]. High levels of serum HMGB1 have been observed in SLE patients and these levels correlated with disease activity [186]. HMGB1-DNA complexes participate in the pathology of SLE and immunization of mice with HMGB1-DNA complexes stimulated production of anti-DNA antibodies, leading to SLE pathology [187]. HMGB1-DNA complexes also trigger the production of type I IFN, a crucial factor in the development of SLE [187,188]. A positive correlation was observed between HMGB1 and pro-inflammatory cytokines such as IL-6 and TNF-α in SLE patients, indicating that HMGB1 might be a prognostic factor for SLE [186]. Administration of an HMGB1 antagonist alleviated the disease in experimental lupus models [189,190].

In addition, extracellular HMGB1 was shown to be a source of NET as evidence that HMGB1 levels was high to greater certain among lupus nephritis [191]. HMGB1 provoked self-DNA-induced macrophage activation by facilitating DNA accumulation in endosomes, which was involved in the development of lupus nephritis [192]. Inversely, inhibition of HMGB1 attenuated the severity of SLE, providing an attractive target for SLE treatment.

### 5.3. Acute Liver Injury

Acute liver injury is a common condition caused by, for example, hepatic viral infection, drug-induced liver dysfunction, or ischemia/reperfusion injury [193]. Patients with liver failure show elevated serum concentrations of HMGB1 [194].

Acetaminophen is the most common cause of drug-induced liver injury. A hepatotoxic intracellular acetaminophen metabolite causes hepatocyte necrotic death, release of non-acetylated HMGB1 and leukocyte activation. Both the release of HMGB1 and the accompanying activation of macrophages contribute to the inflammatory environment of acetaminophen-induced hepatotoxicity [194,195,196]. These processes trigger a second wave of HMGB1, which induces fulminant inflammatory reactions [197]. Acetaminophen-induced hepatic dysfunction is associated with elevated levels of circulating HMGB1, sRAGE and a newly identified extracellular RAGE-binding protein [196]. These molecules function as biomarkers of systemic inflammatory reactions and severity of liver injury. In a mouse model of acetaminophen overdose, HMGB1 antagonists mitigated the negative outcomes, which may be a useful treatment strategy [32].

Serum HMGB1 is also recognized as an early clinical marker of hepatic ischemia/reperfusion injury. HMGB1-associated partners such as TLR4, RAGE and CXCR4 are abundantly expressed in various liver and immune cells, including sinusoidal endothelial cells, stellate cells, hepatocytes, Kupffer cells and dendritic cells [193]. Reducing levels of secreted HMGB1 effectively attenuates hepatic damage and improves survival in mice [32].

### 5.4. Lung Diseases

Extracellular HMGB1 has been postulated to act as a danger signal that evokes an inflammatory storm, prevents the phagocytosis of apoptotic cells (necessary to stop inflammation) and changes vascular remodeling in various lung diseases [13,198].

Asthma is a clinical syndrome characterized by three distinct components: airway inflammation, exaggerated bronchoconstrictor responses and recurrent episodes of airway obstruction [34,199]. HMGB1 was shown to strongly stimulate the differentiation of Th cells and innate lymphocytes (e.g., DCs, group 2 innate lymphoid cells (ILC2s)) and contribute to asthma [128,135,136,148,200]. Levels of HMGB1 in the airway are elevated in animal models and patients with asthma. In severe asthmatic patients, the elevation of airway HMGB1 is associated with an increase in pro-inflammatory cytokines, chemokines, matrix metalloproteinase and counts of blood neutrophils, as well as with activation of neutrophils [201]. HMGB1 activated ILC2s and DCs in a mouse model of asthma, exacerbating the disease [202,203].

Targeting HMGB1 may a useful treatment for asthma. HMGB1 antagonists repressed leukocyte infiltration, levels of collagen, cell counts and pro-inflammatory cytokines in animal models, which attenuated airway remodeling and airway hyperresponsiveness [204,205,206].

Acute lung injury (ALI) is characterized by sudden-onset, severe impairment of pulmonary gas exchange and sustained lung inflammation and it is most frequently caused by infections, sepsis, trauma, or inhalation of toxic substances [207]. The role of HMGB1 in ALI has been explored in animal models. In analogy to endotoxin-induced ALI, intratracheal administration of HMGB1 stimulated infiltration by interstitial/intra-alveolar neutrophils and led to increased alveolar capillary permeability and lung edema in mice [26,94,208,209,210]. During pulmonary infection, HMGB1 is released into the airways by immune cells as well as by damaged cells, where it promotes activation of various immune cells via pattern recognition receptor (PRR) signaling. HMGB1 synergizes with macrophages, neutrophils and pneumocytes to accelerate the production of numerous pro-inflammatory mediators and exacerbate the inflammatory response [65,94,105,127,160]. HMGB1 also compromises endothelial junctions and alveolar capillary permeability, leading to dysfunction of the alveolar endothelial/epithelial barrier [207,211]. HMGB1 levels even correlate positively with pneumonia severity in patients [212].

Intratracheal administration of anti-HMGB1 antibodies alleviate lung inflammation, reduce bacterial burden and improve mortality in mouse models of ALI [213,214,215]. Ethyl pyruvate, 2-O,3-O desulfated heparin and curcumin suppress the release of HMGB1 ameliorating inflammation-induced lung injury [96,114,115,216].

### 5.5. Cardiac Injury

HMGB1 performs numerous functions during cardiac injury, cardiac remodeling and regeneration [217,218]. It contributes to cardiac inflammatory injury by inducing cardiomyocyte senescence, apoptosis, necroptosis and necrosis [35].

HMGB1 is released by immune cells or stressed cardiomyocytes and it recruits Ly6C+ monocytes to the damaged cardiac tissue, thereby reprogramming the monocytes into pro-inflammatory M1 macrophages [217]. M1 macrophages activate expansion of CD4+ T cells, which drive the pathogenesis of myocarditis [218]. Likewise, HMGB1 can activate Th17 cell expansion, subsequent production of IL-17 and recruitment of infiltrating neutrophils, thereby resulting in cardiac injury. Additionally, HMGB1 acts as an autophagy sensor [217]. ROS-induced HMGB1 release from the nucleus activates autophagic flux, which is deleterious for myocytes [218]. HMGB1 also prevents macrophages from phagocytosing apoptotic cells to resolve inflammation [66,101,108,122].

At the same time, HMGB1 may drive lymphocyte apoptosis, lymphocyte homing, inflammatory resolution and heart remodeling [122,149]. Cardiac stress or injury upregulates a novel family of innate lymphoid cells (ILCs), which limit inflammation. All-thiol or disulphide HMGB1 can expand group 3 innate lymphoid cells (ILC3), promoting IL-22 release and attenuating experimental autoimmune myocarditis [8]. In mice, silencing of HMGB1 mitigates experimental autoimmune myocarditis [10]. Similarly, Zhou et al. observed that the cardioprotective effects of HMGB1 prior to ischemia-reperfusion could be regulated by enhancement of vascular endothelial growth factor expression and PI3K/Akt signaling [219]. Overall, the molecular mechanisms underlying the beneficial and detrimental effects of HMGB1 on cardiac injury remain to be elucidated.

### 5.6. Encephalopathy

Encephalopathy is a severe condition caused by sepsis, metabolic diseases and liver dysfunction. HMGB1 can trigger inflammatory cascades in the central nerve system (CNS) and cause astrocyte-mediated cerebral swelling [220].

Up to 25% of septic patients display cognitive decline and animal models of sepsis show significant chronic impairments in memory and learning, which are associated with anatomical changes in the hippocampus [221]. Administration of HMGB1 to septic mice impairs their learning and memory [221] and HMGB1 has been reported to mediate neuroinflammation by inducing apoptosis and local inflammation as well as by compromising blood–brain barrier integrity [222]. HMGB1 may directly potentiate hippocampal inflammatory responses and activate microglia. Suppression of HMGB1 accelerated recovery of neurological function by shifting microglia to the anti-inflammatory M2 phenotype and away from the pro-inflammatory M1 phenotype [111,113]. HMGB1 in macrophages/microglia may be a promising target for treating brain injury. For example, the HMGB1 inhibitor Icariin can prevent LPS-mediated neuroinflammation [223]. Of interest, HMGB1 played detrimental and beneficial role in inflammation and recovery following stroke. During the acute phase, HMGB1 could trigger influx of damaging inflammatory cells and necrosis. However, HMGB1 exhibited beneficial effects in recovery of neurovascular unit during the delayed phase [224]. Therefore, it was proposed that HMGB1-mediated inflammation could be biphasic actions, depending on varied cellular contexts.

### 5.7. Other Inflammatory Conditions

Severe trauma is characterized by extreme cellular stress and excess release of alarmins, especially HMGB1. Indeed, experimental and clinical studies have implicated HMGB1 in systemic inflammation and multiple organ dysfunction as a result of severe trauma [11,225]. In trauma patients, there is an initial plasma HMGB1 peak, exponential decay, then a second HMGB1 wave, peaking at 3–6 h after the trauma [226]. In that study, hyperacetylated and active disulfide HMGB1 isoform appeared in the second wave but not the first one. Neutralization of HMGB1 reduces inflammatory responses and favors survival of animals exposed to trauma, so the biphasic release implies a relatively long therapeutic window [11,226].

HMGB1 plays a critical role in bacterial translocation in the gut and in systemic inflammation during severe acute pancreatitis (SAP). Serum HMGB1 levels are elevated in patients with SAP [227]. In these patients, upregulation of circulating HMGB1 correlates with activation of autophagy and necrosis but with a decrease in plasma sRAGE [112]. During the early phase of acute pancreatitis, HMGB1 may stimulate pancreatic pain by targeting CXCL12/CXCR4 and triggering RAGE signaling [112].

HMGB1 has been implicated in tumorigenesis, progression, metastasis and chemotherapy resistance of various cancers [228,229,230]. HMGB1 could elicit proinflammatory cascades and promote formation and metastasis of tumor, which is crucial for sustenance of tumor inflammatory microenvironment [228]. In addition, HMGB1 was critical for the activation and intratumoral aggregation of tumor-infiltrating T cells to release lymphotoxin α1β2 and attract macrophages into the lesion, thereby accelerating tumor progression [229]. Moreover, HMGB1 was observed to incite apoptosis in macrophage-derived dendritic cells, subsequently dampening anti-tumor immunity [230]. More recently, Haruna et al. reported that docetaxel could engage in anti-tumor immunity via HMGB1 release, implying a beneficial role of HMGB1 in anti-cancer treatment [231]. However, the exact mechanism of the effect needs to be investigated in future study.

**Table 1 cells-10-01044-t001:** Summary of studies concerning the significance of HMGB1 in various inflammatory diseases.

Diseases	Year	Authors	Observations or Conclusions	Ref.
Sepsis	1999	Wang etal.	HMGB1 acts as a late meditator of endotoxin lethality in mice.	[77]
	2010	Lamkanfi et al.	HMGB1 release critical for endotoxin occurs downstream of inflammasome assembly and caspase 1 activation.	[78]
	2015	Hwang et al.	Theacetylation-dependent interaction between HMGB1 and SIRT1 is critical for LPS-induced lethality in an experimental model of sepsis.	[164]
	2011	Youn et al.	HMGB1 has two LPS-binding peptide regions that can be utilized to design anti-sepsis or LPS-neutralizing therapeutics in a mouse model.	[165]
	2004	Yang et al.	Specific inhibition of HMGB1 protects against the development of organ injury and increases survival in septic mice.	[166]
	2016	Valdes-Ferrer et al.	HMGB1 mediates anemia of inflammation by interfering with erythropoiesis in murine sepsis survivals.	[167]
	2013	Valdes-Ferrer et al.	HMGB1 mediates splenomegaly and expansion of splenic CD11b+ly-6C(high) inflammatory monocytes in murine sepsis survivors.	[168]
	2017	Stevens et al.	Anti-HMGB1 antibodies alter inflammation in murine sepsis model and reduce sepsis mortality without potentiating immunosuppression.	[169]
	2017	Gregoire et al.	HMGB1 induces neutrophil dysfunction in septic mice and in patients who survive septic shock.	[170]
	2016	Gil et al.	Naringin reduces the release of TNF-α and HMGB1 from LPS-stimulated macrophages and improves survival in a CLP-induced sepsis mice.	[171]
	2006	Suda et al.	Anti-HMGB1 antibodies improve survival of rats with sepsis.	[172]
Arthritis	2003	Taniguchi et al.	HMGB1 is strongly expressed in synovial fluid of RA patients inducing the release of proinflammatory cytokine from synovial fluid macrophages.	[173]
	2007	Goldstein et al.	Cholinergic anti-inflammatory pathway activity and HMGB1 serum levels in patients with RA.	[175]
	2010	Ostberg et al.	HMGB1 is involved in the pathogenesis of this spontaneous polyarthritis and intervention with an HMGB1 antagonist can mediate beneficial effects.	[177]
	2011	Schierbeck et al.	Monoclonal anti-HMGB1 antibody significantly ameliorates the clinical courses and partially prevents joint destruction in collagen type II-induced arthritis and spontaneous arthritis model.	[178]
	2003	Pullerits et al.	HMGB1 triggers joint inflammation by activating macrophages and inducing production of IL-1 via NF-κB activation.	[179]
	2008	Hamada et al.	HMGB1 is a coupling factor for hypoxia and inflammation in arthritis.	[180]
	2016	Lundback et al.	HMGB1 is synovial fluid from idiopathic arthritis patients actively released through both acetylation-dependent and nondependent manners.	[181]
	2003	Kokkola et al.	Successful treatment of collagen-induced arthritis in mice and rats by targeting extracellular HMG1 activity.	[182]
	2006	Wouwer et al.	The lectin-like domain of thrombomodulin interferes with complement activation and protects against arthritis in mouse model.	[183]
	2008	Zetterstrom et al.	Gold sodium thiomalate inhibits the extracellular release of HMGB1 from activated macrophages and causes the nuclear retention of this protein, suggesting the anti-rheumatic effects of gold sodium thiomalate in RA.	[184]
SLE	2012	Ma et al.	Elevated plasma level of HMGB1 is associated with disease activity and combined alterations with IFN-α and TNF-α in SLE.	[186]
	2011	Abdulahad et al.	Levels of HMGB1 in the sera of SLE patients, in particular in those with active renal disease, are increased. Serum HMGB1 levels are related to SLE disease activity index scores and proteinuria, as well as to levels of anti-HMGB1 antibodies.	[189]
	2014	Zhang et al.	HMGB1 inhibition attenuates lupus-like disease in BXSB mice.	[190]
	2019	Whittall-Garcia et al.	In SLE patients, NETs are a source of extracellular HMGB1, which correlates with clinical and histopathological features of active nephritis.	[191]
	2015	Li et al.	Extracellular HMGB1 facilitates self-DNA induced macrophage activation via promoting DNA accumulation in endosomes and contributes to the pathogenesis of lupus nephritis.	[192]
Liver injury	2010	Evankovich et al.	High mobility group box 1 release from hepatocytes during ischemia and reperfusion injury is mediated by decreased histonedeacetylase activity.	[193]
	2016	Lundback et al.	Anti-HMGB1 polyclonal antibody significantly attenuates serum elevations of alanine aminotransferase and abrogates markers of inflammation and improves survival in a model of acetaminophen-acute liver injury.	[197]
	2011	Dragomir et al.	HMGB1 released by acetaminophen-injured hepatocytes leads to macrophage activation.	[196]
Asthma	2017	Di Candia et al.	HMGB1 is upregulated in the airways in asthma and potentiates airway smooth muscle contraction via TLR4.	[201]
	2015	Cuppari et al.	Sputum HMGB1 is increased in asthmatic children and correlates with asthma severity and inversely with lung function indices.	[203]
	2017	Zhang et al.	Vitamin D reduces inflammatory response in asthmatic mice via HMGB1/TLR4/NF-κB pathway.	[205]
Acute lung injury	2015	Sodhi et al	Intestinal epithelial TLR4 activation leads to HMGB1 release from gut and the development of lung injury.	[208]
	2005	Kim et al.	Hemorrhage results in increased HMGB1 expression in the lung primarily via neutrophil sources.	[210]
	2013	Patel et al.	HMGB1 mediates hyperoxia-induced impairment of *Pseudomonas aeruginosa* clearance and inflammatory lung injury in mice.	[212]
	2014	Entezari et al.	Inhibition of extracellular HMGB1 attenuates hyperoxia-induced inflammatory acute lung injury.	[214]
	2004	Ueno et al.	HMGB1 is increased in plasma and lung epitheliallining fluid of patients with acute lung injury and mice instilled with lipopolysaccharide.	[215]
Cardiac injury	2009	Kohno et al.	Elevated serum HMGB1 of is associated with adverse clinical outcomes in patients with myocardial infarction. HMGB1 blockade intramyocardial infarction model aggravated left ventricular remodeling possibly via impairment of the infarct-healing process.	[217]
	2008	Kitahara et al.	HMGB1 enhances angiogenesis, restores cardiac function and improves survival after myocardial infarction in mice.	[218]
Encepha-lopathy	2014	Zou et al.	Ethanol alters histone deacetylases that regulate HMGB1 release and that danger signal HMGB1 as endogenous ligand for TLR4 mediates ethanol-induced brain neuroimmune signaling via activation of microglial TLR4.	[220]
	2012	Chavan et al.	Elevated HMGB1 mediates cognitive decline in sepsis survivors in mice.	[221]
	2019	Liu et al.	Icariin and icaritin ameliorate hippocampus neuroinflammation via inhibiting HMGB1-related pro-inflammatory signals in lipopolysaccharide-induced inflammation model in mice.	[223]
Trauma	2007	Levy et al.	HMGB1 levels are transiently elevated just 1 h after injury in both wild-type and TLR4 mutant mice.	[225]
	2012	Shimazaki et al.	Anti-HMGB1 antibody reduces inflammatory reactions and improve survival via blocking extracellular HMGB1 in a rat model of crush injury.	[226]

## 6. Conclusions and Perspectives

HMGB1 is a critical endogenous nucleoprotein with multiple biological functions. The properties of HMGB1 depend on its subcellular location, interaction partners and post-translational modifications (oxidation, phosphorylation and acetylation). Normally, HMGB1 acts as an architectural factor that modulates nucleosomes, DNA and gene transcription by binding DNA in the nucleus. Under stress conditions, HMGB1 can shuttle to the cytoplasm, where it is actively or passively released from cells. HMGB1 is actively released by various types of immune cells, such as neutrophils, macrophages and DCs and it is also secreted during cell injury and death. Once outside the cell, the protein participates in a broad range of biological behaviors, such as autophagy, apoptosis and pyroptosis.

HMGB1 has metabolic, immune, chemokine and cytokine activities. HMGB1 induces and maintains potent inflammatory and immune responses by influencing various immune cells involved in the pathology of inflammatory conditions. HMGB1 can recruit and activate neutrophils, promote neutrophil infiltration and promote NET formation. It modulates maturation, differentiation, activation and apoptosis of DCs. It induces apoptosis and pyroptosis in macrophages while impairing their ability to phagocytose apoptotic cells. HMGB1 promotes the migration and survival of regulatory T cells, while suppressing their activity. HMGB1 can modulate the immune functions of T lymphocytes in a dose-dependent fashion.

Accumulating evidence indicates that HMGB1 represents a potential biomarker and an appealing target for innovative therapeutic approaches in numerous inflammatory disorders, such as sepsis, lung diseases, autoimmune diseases, acute liver injury, cardiac injury, encephalopathy and other inflammatory conditions. Targeting HMGB1 with pharmacological inhibitors and monoclonal antibodies has shown promise for repressing HMGB1-mediated inflammatory responses in preclinical studies. Blocking HMGB1 can mitigate the cytokine storm and thereby alleviate tissue injury and reduce mortality in various animal models of multiple inflammatory disorders.

Targeting HMGB1 has yet to be translated to the clinic. We need further exploration into how HMGB1 drives the pathogenesis of several inflammatory conditions. Considering the dose-dependent effects of HMGB1 in certain contexts, standardized, accurate measurement of HMGB1 levels is key to explore therapeutic strategies based on HMGB1 antagonists. Most preclinical studies have focused on the significance of extracellular HMGB1, so more work is needed on the functions of intracellular HMGB1 and its targetability for the treatment of inflammatory diseases.

## Figures and Tables

**Figure 1 cells-10-01044-f001:**
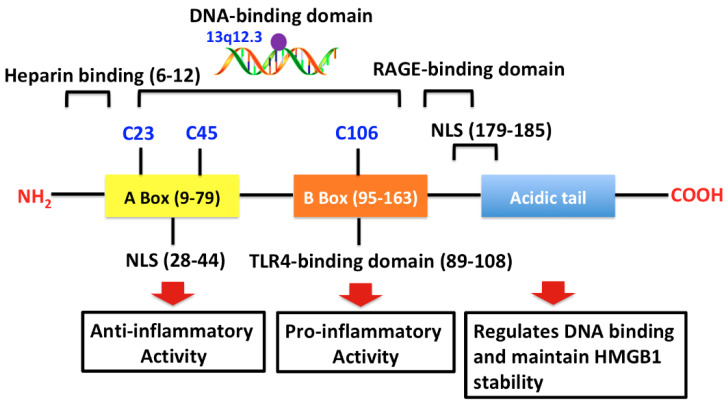
Structure of the HMGB1 protein. The HMGB1 protein comprises 215 amino acid residues and contains three regions: A-box, B-box and an acidic tail. Each region includes several functional domains: NLS, TLR4-binding domain, heparin-binding, RAGE-binding domain and DNA-binding domain. The domains exert different biological functions, such as anti-inflammatory activity, pro-inflammatory activity, regulation of DNA binding and stabilization of HMGB1. The A box mediates the anti-inflammatory functions of HMGB1; the B box, cytokine-mediated functions. Three key cysteines are also indicated (Cys106, Cys45 and Cys23). Abbreviations: HMGB1, high-mobility group box 1; NLS, nuclear localization sites; TLR, Toll-like receptor; RAGE, receptor for advanced glycation end products.

**Figure 2 cells-10-01044-f002:**
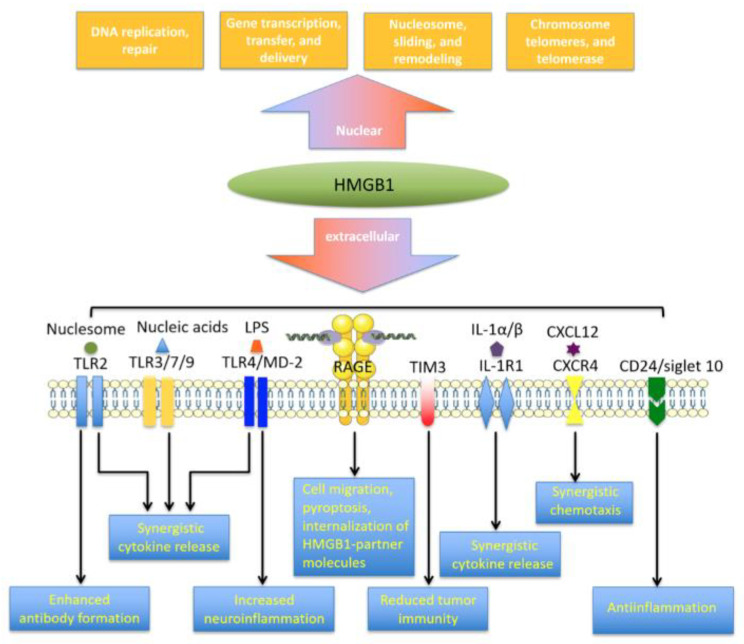
HMGB1 functions and associated receptor families. In the nucleus, HMGB1 functions as an architectural molecule possessing DNA-binding and -bending properties. Extracellular HMGB1, on the other hand, signals via multiple receptors such as TLR4/MD-2, TLR2, TLR3/7/9, RAGE, CD24, singlet 10, integrin/Mac1, TIM3, IL-1R1, or CXCR4. In conjunction with other mediators, HMGB1 induces cytokine release, cell migration, pyroptosis and internalization of HMGB1 binding partners; it recruits cells; it reduces inflammation and tumor immunity; it increases neuroinflammation; and it enhances autoantibody formation. Abbreviations: HMGB1, high-mobility group box 1; IL, interleukin; RAGE, receptor for advanced glycation end products; TLR, Toll-like receptor.

**Figure 3 cells-10-01044-f003:**
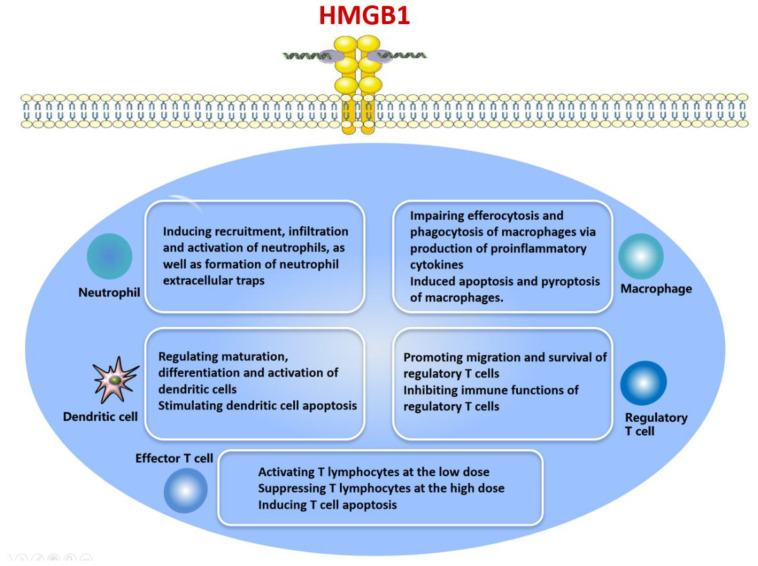
Effects of HMGB1 on different types of immune cell.

**Figure 4 cells-10-01044-f004:**
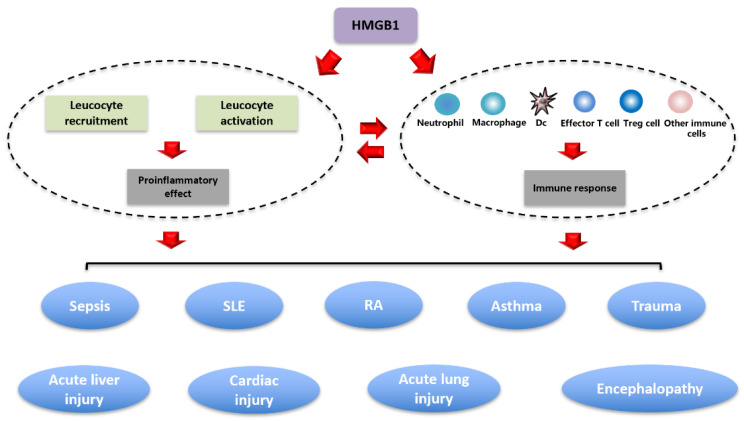
Roles of HMGB1 in various inflammatory disorders. The studies discussed in this review show that increased levels of extracellular HMGB1, whether in serum or tissue, can signal as a DAMP molecule to regulate inflammatory and immune responses, contributing to various inflammatory disorders, such as sepsis, SLE, RA, asthma, trauma, acute liver injury, cardiac injury, acute lung injury and encephalopathy. Abbreviations: HMGB1, high-mobility group box 1; Treg cell, regulatory T cell; DC, dendritic cell; DAMP, damage-associated molecular pattern; RA, rheumatoid arthritis; SLE, systemic lupus erythematosus. Abbreviations: HMGB1, high-mobility group box 1; RAGE, receptor for advanced glycation end products; TLR, Toll-like receptor; IL, interleukin; Th, T helper; NF-κB, nuclear factor-κB; LPS, lipopolysaccharide; SIRT1, sirtuin 1; TNF-α, tumor necrosis factor α; CLP, cecal ligation and puncture; RA, rheumatoid arthritis; SLE, systemic lupus erythema.

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
