# Peer review of "The Effect and Regulatory Mechanism of High Mobility Group Box-1 Protein on Immune Cells in Inflammatory Diseases"

_cells, 2021, doi:10.3390/cells10051044_

Round 1
Reviewer 1 Report
In this manuscript, Ge et al. discuss the role of HMGB1 protein on immune cells in inflammatory diseases. It is a comprehensive description of the topic with characteristics of the protein followed by analysis of the HMGB1 receptor network, effects of the protein on immune cells and its role in inflammatory disorders. The authors thoroughly discuss the critical role of this molecule with multiple biological functions. This review provides a solid amount of data on the topic and provides an overview of the important and modern pieces of literature.
My general impression is that this is a review of an important area of research with emerging clinical implications. The authors should be applauded for inclusion of the most important facts and an interesting narration. However, several drawbacks should be pointed out.
Major issues:
- HMGB1 clearly is a double edged sword, as mentioned by the authors. Detrimental role of this molecule in many clinical situations was described extensively. However, less studied beneficial role of HMGB1 is given little attention, yet it may be very relevant clinically (see the analogy with NO inhibitors which proved detrimental in sepsis because NO plays also important constitutive role). The authors should mention some manuscripts describing beneficial role of HMGB1, e.g. Haruna et al (Docetaxel toxicity), Zhou et al (protection against ischemia-reperfusion) or Hayakawa et al (protection in stroke).
- Line 404, Figure 4: the distinction between inflammatory response and immune response is not clear to me, since according to this Figure they both affect the same list of processes and diseases. This distinction is important because it is even mentioned in the title of the manuscript.
- It is not clear to me what was the basis of selection of the manuscripts included as examples for the Tables, especially Tables 1-5. If the goal was to present the most modern research then why many older manuscripts? If the goal was to present the most significant findings I believe the selection of the manuscripts is not fair and should provide an overview of more worldwide distribution of research on this topic.
Minor issues:
- Please pay more attention to the layout of the Tables; now some of the descriptions are shifted in relation to the authors’ names which makes the text hard to understand.
- Line 178: this short description of MAPK is an oversimplification; MAPK has three well characterized components (Erk, p38 and JNK). We should be always specific, whether we talk about MAPK in general or any of its components. Similarly lines 338-9 talk about MAPK and Erk while it could be only MAPK in general or Erk as one of its components but never at the same time MAPK as a whole and at the same time one of its components (Erk).
- Line 304: skewing Th17 differentiation is ambiguous; skewing in which direction? Also, HMHG1 in this line seems to be a misspelled word.
- Line 331-6: observation of opposite effects of HMGB1 on T lymphocyte function may herald problems with clinical implementation of HMGB1 directed drugs: the same clinical dose may have opposite effects in various patients (due to individual differences in sensitivity to inhibitor) making drugs difficult to dose properly. This potential problem should be discussed whenever hopes for clinical use of HMGB1 directed drugs is discussed elsewhere in the manuscript.
Author Response
Apr. 2, 2021, Beijing, China
Dear Prof. Libby He and reviewer:
Thank you very much for your kind letter dated on Mar. 26, 2021, and sending us the information concerning our revised manuscript "The effect and its regulatory mechanism of high mobility group box-1 protein on immune cells in inflammatory diseases". Enclosed please find our re-revised version with some changes according to the reviewer’s suggestions.
We greatly appreciate the reviewer’s suggestions. Those suggestions are all valuable and very helpful for revising and improving our paper. We have studied comments carefully and have revised the manuscript which we hope meet with approval. We would like to take the opportunity to point out as follows.
Reviewer
In this manuscript, Ge et al. discuss the role of HMGB1 protein on immune cells in inflammatory diseases. It is a comprehensive description of the topic with characteristics of the protein followed by analysis of the HMGB1 receptor network, effects of the protein on immune cells and its role in inflammatory disorders. The authors thoroughly discuss the critical role of this molecule with multiple biological functions. This review provides a solid amount of data on the topic and provides an overview of the important and modern pieces of literature. My general impression is that this is a review of an important area of research with emerging clinical implications. The authors should be applauded for inclusion of the most important facts and an interesting narration. However, several drawbacks should be pointed out.
Major issues:
- HMGB1 clearly is a double edged sword, as mentioned by the authors. Detrimental role of this molecule in many clinical situations was described extensively. However, less studied beneficial role of HMGB1 is given little attention, yet it may be very relevant clinically (see the analogy with NO inhibitors which proved detrimental in sepsis because NO plays also important constitutive role). The authors should mention some manuscripts describing beneficial role of HMGB1, e.g. Haruna et al (Docetaxel toxicity), Zhou et al (protection against ischemia-reperfusion) or Hayakawa et al (protection in stroke).
Response: We greatly appreciate the reviewer’s comments and valuable suggestions. We included relevant information in the revised version.
“Similarly, Zhou et al. observed that the cardioprotective effects of HMGB1 prior to I/R could be regulated by enhancement of vascular endothelial growth factor expression and PI3K/Akt signaling [219]. Overall, the molecular mechanisms underlying the beneficial and detrimental effects of HMGB1 on cardiac injury remain to be elucidated.” (lines 526~531, highlighted in red).
“Of interest, HMGB1 played detrimental and beneficial role in inflammation and recovery following stroke. During the acute phase, HMGB1 could trigger influx of damaging inflammatory cells and necrosis. However, HMGB1 exhibited beneficial effects in recovery of neurovascular unit during the delayed phase [224]. Therefore, it was proposed that HMGB1-mediated inflammation could be biphasic actions, depending on varied cellular contexts.” (lines 546~552, highlighted in red).
“HMGB1 could elicit proinflammatory cascades, and promote formation and metastasis of tumor, which is crucial for sustenance of tumor inflammatory microenvironment [228]. Besides, HMGB1 was critical for the activation and intratumoral aggregation of tumor-infiltrating T cells to release lymphotoxin α1β2 and attract macrophages into the lesion, thereby accelerating tumor progression [229]. Moreover, HMGB1 was observed to incite apoptosis in macrophage-derived dendritic cells, subsequently dampening anti-tumor immunity [230]. More recently, Haruna et al. reported that docetaxel could engage in anti-tumor immunity via HMGB1 release, implying a beneficial role of HMGB1 in anti-cancer treatment [231]. However, the exact mechanism of the effect needs be investigated in future study.” (lines 570~580, highlighted in red)
- Line 404, Figure 4: the distinction between inflammatory response and immune response is not clear to me, since according to this Figure they both affect the same list of processes and diseases. This distinction is important because it is even mentioned in the title of the manuscript.
Response: We greatly appreciate for reviewer's advices. We made relevant corrections in Figure 4 in the revised version. (Figure 4)
- It is not clear to me what was the basis of selection of the manuscripts included as examples for the Tables, especially Tables 1-5. If the goal was to present the most modern research then why many older manuscripts? If the goal was to present the most significant findings I believe the selection of the manuscripts is not fair and should provide an overview of more worldwide distribution of research on this topic.
Response: Thanks for reviewer's suggestions and careful works. We deleted Table 1-5 and made some corrections in Table 6 in the revised version.
Minor issues:
- Please pay more attention to the layout of the Tables; now some of the descriptions are shifted in relation to the authors’ names which makes the text hard to understand.
Response: We greatly appreciate for reviewer' advices. We made relevant corrections in the revised version.
- Line 178: this short description of MAPK is an oversimplification; MAPK has three well characterized components (Erk, p38 and JNK). We should be always specific, whether we talk about MAPK in general or any of its components. Similarly lines 338-9 talk about MAPK and Erk while it could be only MAPK in general or Erk as one of its components but never at the same time MAPK as a whole and at the same time one of its components (Erk).
Response: Thanks for the reviewer's comments. We made some corrections in the revised version.
“Binding of extracellular HMGB1 to TLRs and RAGE on the surface of immune cells initiates downstream signaling cascades involving extracellular signal-regulated kinase 1/2 (ERK1/2), p38 mitogen-activated protein kinase (MAPK), phosphoinositide-3-kinase (PI3K)/RAC-α serine/threonine-protein kinase (Akt), and NF-κB (p56) [24,65,81,84-87].” (lines 178~181, highlighted in red)
"Hp91-mediated DC activation is dependent on TLR4, MyD88, and IFNαβR, and it is mediated by NF-κB and p38 MAPK cascades [131,132].” (lines 277~278, highlighted in red)
“IL-2 potently stimulates T lymphocytes and regulates the balance between Th1 and Th2 cells. HMGB1 exerts its immunosuppressive effect on T lymphocytes via p53, p38 MAPK and ERK 1/2 [143-149]. p38 MAPK and ERK1/2 may act on the transcription factor NF-AT in HMGB1-induced immunosuppression.” (lines 325~328, highlighted in red).
- Line 304: skewing Th17 differentiation is ambiguous; skewing in which direction? Also, HMHG1 in this line seems to be a misspelled word.
Response: We greatly appreciate for reviewer' advices. We made some corrections in the revised version.
“HMGB1-dependent DCs activation induces Th17-type responses, and this activation can be blocked using a soluble form of RAGE (sRAGE) [136], which blocks interaction between HMGB1 and endogenous RAGE."(lines 295~297, highlighted in red).
- Line 331-6: observation of opposite effects of HMGB1 on T lymphocyte function may herald problems with clinical implementation of HMGB1 directed drugs: the same clinical dose may have opposite effects in various patients (due to individual differences in sensitivity to inhibitor) making drugs difficult to dose properly. This potential problem should be discussed whenever hopes for clinical use of HMGB1 directed drugs is discussed elsewhere in the manuscript.
Response: Thanks for the reviewer's comments. We added relevant information in the revised manuscript.
“Considering the dose-dependent effects of HMGB1 in certain contexts, standardized, accurate measurement of HMGB1 levels is key to explore therapeutic strategies based on HMGB1 antagonists.” (lines 611~614, highlighted in red).
We greatly appreciate for editor/reviewers' warm works earnestly, and hope the version of our manuscript will meet with approval. Once again, thank you very much for your consideration of this paper.
Best regards.
Yours sincerely,
Yong-ming Yao, M.D, Ph.D.
Translational Medicine Research Center, Medical Innovation Research Division and Fourth Medical Center of the Chinese PLA General Hospital,
51 Fu-Cheng Road, Beijing 100048,
People's Republic of China.
Tel: (+86)1066867394,
Fax: (+86)1068989158,
E-mail: c_ff@sina.com
Reviewer 2 Report
This review manuscript is generally well described and contained the subtopics to understand for readers. However, some or many cited references for scientific information are not carefully selected from the originally published papers, but from the other review papers or not original papers. The authors need to correct this and recommend careful reading required. And one minor thing is the text alignment in all the tables need to correction.
Author Response
Apr. 2, 2021, Beijing, China
Dear Prof. Libby He and reviewer:
Thank you very much for your kind letter dated on Mar. 26, 2021, and sending us the information concerning our revised manuscript "The effect and its regulatory mechanism of high mobility group box-1 protein on immune cells in inflammatory diseases". Enclosed please find our re-revised version with some changes according to the reviewer’s suggestions.
We greatly appreciate the reviewer’s suggestions. Those suggestions are all valuable and very helpful for revising and improving our paper. We have studied comments carefully and have revised the manuscript which we hope meet with approval. We would like to take the opportunity to point out as follows.
Reviewer 2
This review manuscript is generally well described and contained the subtopics to understand for readers. However, some or many cited references for scientific information are not carefully selected from the originally published papers, but from the other review papers or not original papers. The authors need to correct this and recommend careful reading required. And one minor thing is the text alignment in all the tables need to correction.
Response: Thanks for the reviewer's comments. These suggestions are all valuable and very helpful for revising and improving our paper. We made some corrections in the revised version according to the reviewer’s recommendations. (highlighted in red in the revised version).
We greatly appreciate for editor/reviewers' warm works earnestly, and hope the version of our manuscript will meet with approval. Once again, thank you very much for your consideration of this paper.
Best regards.
Yours sincerely,
Yong-ming Yao, M.D, Ph.D.
Translational Medicine Research Center, Medical Innovation Research Division and Fourth Medical Center of the Chinese PLA General Hospital,
51 Fu-Cheng Road, Beijing 100048,
People's Republic of China.
Tel: (+86)1066867394,
Fax: (+86)1068989158,
E-mail: c_ff@sina.com
Reviewer 3 Report
Manuscript ID: cells-1144839
Type of manuscript: Review
Title: The effect and its regulatory mechanism of high mobility 2 group box-1 protein on immune cells in inflammatory diseases
Authors Yun Ge, Man Huang and Yong-ming Yao
The reviewed manuscript by Yun Ge et al. is focused on the effects of HMGB1 on various immune cell types and describes the molecular mechanisms by which HMGB1 contributes to the development of inflammatory disorders. The subject would be of interest to the scientific community, and the present manuscript provides a clear and comprehensive summary of the recent literature, including the own papers of the authors. After a brief description of the main HMGB1 characteristics, the authors present evidence regarding the effects of HMGB1 on immune cells (neutrophils, macrophages, dendritic cells, T lymphocytes, regulatory T cells) and its role in inflammatory disorders during sepsis, autoimmune and lung diseases, cardiac injury and encephalopathy. The review is well written. The points are clearly presented and the conclusions are correct.
Minor point:
- Page 8, line 268: native instead naive T cells
- Page 14: Table 6 lacks the titles of the last two references (223 and 224) in the trauma section
- I ask the authors to edit the journal abbreviations in the reference section (e.g. they are somewhere with dots, somewhere without; reference 223 is without abbreviation for American Journal of Physiology).
Author Response
Apr. 2, 2021, Beijing, China
Dear Prof. Libby He and reviewer:
Thank you very much for your kind letter dated on Mar. 26, 2021, and sending us the information concerning our revised manuscript "The effect and its regulatory mechanism of high mobility group box-1 protein on immune cells in inflammatory diseases". Enclosed please find our re-revised version with some changes according to the reviewer’s suggestions.
We greatly appreciate the reviewer’s suggestions. Those suggestions are all valuable and very helpful for revising and improving our paper. We have studied comments carefully and have revised the manuscript which we hope meet with approval. We would like to take the opportunity to point out as follows.
Reviewer 3
The reviewed manuscript by Yun Ge et al. is focused on the effects of HMGB1 on various immune cell types and describes the molecular mechanisms by which HMGB1 contributes to the development of inflammatory disorders. The subject would be of interest to the scientific community, and the present manuscript provides a clear and comprehensive summary of the recent literature, including the own papers of the authors. After a brief description of the main HMGB1 characteristics, the authors present evidence regarding the effects of HMGB1 on immune cells (neutrophils, macrophages, dendritic cells, T lymphocytes, regulatory T cells) and its role in inflammatory disorders during sepsis, autoimmune and lung diseases, cardiac injury and encephalopathy. The review is well written. The points are clearly presented and the conclusions are correct.
Response: Many thanks for reviewer's comments.
Minor point:
- Page 8, line 268: native instead naive T cells
Response: Thanks for the reviewer's comments. We made some corrections in the revised manuscript.
“Dendritic cells (DCs) serve as professional antigen-presenting cells that capture antigen in one location, then migrate to lymph nodes, where they present it to native T cells, initiating the adaptive immune response [127].” (lines 263~265, highlighted in red)
- Page 14: Table 6 lacks the titles of the last two references (223 and 224) in the trauma section
Response: Thanks for the reviewer's advices. We included relevant information in the table in the revised version.
- I ask the authors to edit the journal abbreviations in the reference section (e.g. they are somewhere with dots, somewhere without; reference 223 is without abbreviation for American Journal of Physiology).
Response: Thanks for the reviewer's recommendations. We had revised all the mistakes.
We greatly appreciate for editor/reviewers' warm works earnestly, and hope the version of our manuscript will meet with approval. Once again, thank you very much for your consideration of this paper.
Best regards.
Yours sincerely,
Yong-ming Yao, M.D, Ph.D.
Translational Medicine Research Center, Medical Innovation Research Division and Fourth Medical Center of the Chinese PLA General Hospital,
51 Fu-Cheng Road, Beijing 100048,
People's Republic of China.
Tel: (+86)1066867394,
Fax: (+86)1068989158,
E-mail: c_ff@sina.com
Reviewer 4 Report
As a reviewer of numerous articles on HMGB1, I always enjoy a succinct overview of the subject matter as there is some degree of contrasting findings/ideas. This review article covers most of the basics but really none of the controversies. Moreover, many of the citations present in the document are review articles and not the original citations. The use of tables to summarize the conclusions of numerous original papers on the topic can bring to light the enormity of the literature but does nothing to point out the differences.
Author Response
Apr. 2, 2021, Beijing, China
Dear Prof. Libby He and reviewer:
Thank you very much for your kind letter dated on Mar. 26, 2021, and sending us the information concerning our revised manuscript "The effect and its regulatory mechanism of high mobility group box-1 protein on immune cells in inflammatory diseases". Enclosed please find our re-revised version with some changes according to the reviewer’s suggestions.
We greatly appreciate the reviewer’s suggestions. Those suggestions are all valuable and very helpful for revising and improving our paper. We have studied comments carefully and have revised the manuscript which we hope meet with approval. We would like to take the opportunity to point out as follows.
Reviewer 4
As a reviewer of numerous articles on HMGB1, I always enjoy a succinct overview of the subject matter as there is some degree of contrasting findings/ideas. This review article covers most of the basics but really none of the controversies. Moreover, many of the citations present in the document are review articles and not the original citations. The use of tables to summarize the conclusions of numerous original papers on the topic can bring to light the enormity of the literature but does nothing to point out the differences.
Response: Thanks for the reviewer's comments. These suggestions are all valuable and very helpful for revising and improving our paper. In the revised version, we had made some corrections and tried to provide the different effects of HMGB1 on various diseases or states according to the reviewer’s advices. (highlighted in red, e.g., lines 526-531; lines 546-552; lines 570-580)
We greatly appreciate for editor/reviewers' warm works earnestly, and hope the version of our manuscript will meet with approval. Once again, thank you very much for your consideration of this paper.
Best regards.
Yours sincerely,
Yong-ming Yao, M.D, Ph.D.
Translational Medicine Research Center, Medical Innovation Research Division and Fourth Medical Center of the Chinese PLA General Hospital,
51 Fu-Cheng Road, Beijing 100048,
People's Republic of China.
Tel: (+86)1066867394,
Fax: (+86)1068989158,
E-mail: c_ff@sina.com
Round 2
Reviewer 2 Report
Generally this manuscript is well organized and improved.
Author Response
We greatly appreciate the reviewer’s comments and valuable suggestions.
Reviewer 4 Report
The authors herein have had an opportunity to revise this manuscript. My initial criticism of the review article was that while the MS covers most of the basics but none of the controversies. The authors had added minor additions regarding the different interpretations. However, the authors chose not to replace the many of the citations present in the document which are review articles with original citations.
Author Response
Dear Prof. Libby He and reviewers:
Thank you very much for your kind letter dated on Apr. 15, 2021, and sending us the information concerning our revised manuscript "The effect and its regulatory mechanism of high mobility group box-1 protein on immune cells in inflammatory diseases". Enclosed please find our re-revised version with some changes according to the reviewer’s suggestions.
We greatly appreciate the reviewer’s comments and suggestions. Those suggestions are all valuable and very helpful for revising and improving our paper. We have studied comments carefully and have revised the manuscript which we hope meet with approval. We would like to take the opportunity to point out as follows.
Reviewer
The authors herein have had an opportunity to revise this manuscript. My initial criticism of the review article was that while the MS covers most of the basics but none of the controversies. The authors had added minor additions regarding the different interpretations. However, the authors chose not to replace the many of the citations present in the document which are review articles with original citations.
Response: We greatly appreciate the reviewer’s comments and valuable suggestions. We have replaced relevant citations in the revised version (references, e.g., 2, 3, 4, 5, 7, 9, 10, 12, 17, 18, 25, 26, 30, 34, 47, 49, 51, 57, 62, 69, 70, 84, 87, 187, 188, 195, 204, 208, 209, 211, 213, 222, all highlighted in red).
We greatly appreciate for editor/reviewers' warm works earnestly, and hope the version of our manuscript will meet with approval. Once again, thank you very much for your consideration of this paper.
Best regards.
Yours sincerely,
Yong-ming Yao, M.D, Ph.D.
Translational Medicine Research Center, Medical Innovation Research Division and Fourth Medical Center of the Chinese PLA General Hospital,
51 Fu-Cheng Road, Beijing 100048,
People's Republic of China.
Tel: (+86)1066867394,
Fax: (+86)1068989158,
E-mail: c_ff@sina.com